# Signals Orchestrating Peripheral Nerve Repair

**DOI:** 10.3390/cells9081768

**Published:** 2020-07-24

**Authors:** Michela Rigoni, Samuele Negro

**Affiliations:** 1Department of Biomedical Sciences, University of Padua, 35131 Padua, Italy; samuele.negro1987@gmail.com; 2Myology Center (Cir-Myo), University of Padua, 35129 Padua, Italy

**Keywords:** cell signaling, neurodegeneration, neuromuscular junction, neuroregeneration, Schwann cells

## Abstract

The peripheral nervous system has retained through evolution the capacity to repair and regenerate after assault from a variety of physical, chemical, or biological pathogens. Regeneration relies on the intrinsic abilities of peripheral neurons and on a permissive environment, and it is driven by an intense interplay among neurons, the glia, muscles, the basal lamina, and the immune system. Indeed, extrinsic signals from the milieu of the injury site superimpose on genetic and epigenetic mechanisms to modulate cell intrinsic programs. Here, we will review the main intrinsic and extrinsic mechanisms allowing severed peripheral axons to re-grow, and discuss some alarm mediators and pro-regenerative molecules and pathways involved in the process, highlighting the role of Schwann cells as central hubs coordinating multiple signals. A particular focus will be provided on regeneration at the neuromuscular junction, an ideal model system whose manipulation can contribute to the identification of crucial mediators of nerve re-growth. A brief overview on regeneration at sensory terminals is also included.

## 1. Introduction

The peripheral nervous system (PNS) can regenerate following injury, at variance from the central nervous system (CNS) [1,2,3,4,5,6]. Indeed, enabling axon regeneration after CNS injury remains a major challenge in neurobiology [1]. The regeneration capability of the PNS has been retained in higher vertebrates and refined through evolution, given its essential value for survival, which strictly depends on fully functional neuromuscular and sensory apparatuses. In addition to intrinsic qualities of the neurons themselves, glial-driven changes to the neural environment have a significant impact on the regenerative outcome, and the contribution of Schwann cells (SC) to peripheral nerve repair will be highlighted here. 

A particular focus will be on regeneration of the neuromuscular junction (NMJ), the specialized synapse whose functionality is crucial for locomotion. In addition to the motor axon terminal and the muscle fiber, the NMJ consists of several non-myelinating perisynaptic Schwann cells (PSC) that cover and protect the nerve terminal [7,8]. Perfectly coordinated dynamics among NMJ components assure efficient transmission of action potentials from the motor neuron to the muscle. A precise orchestration of multiple signals leads to NMJ maturation, and then a further fine tuning of these signals maintains NMJ function for a long time [7,8,9]. A decay in the generation and elaboration of signals within the NMJ takes place with aging, and this is at the basis of the decreasing functionality of the aged NMJ [10]. 

NMJ functionality can be undermined by traumatic injuries, neurotoxins, autoimmune antibodies, and by genetic alterations causing severe neuroparalytic syndromes. Neuroparalysis can either be completely reversed or can progress to axonal degeneration and permanent loss of function, such as in amyotrophic lateral sclerosis [11,12]. Intense inter- and intra-cellular signaling also occurs at the NMJ during regeneration, with a major role in the process played by PSC [13,14,15,16], and will be briefly reviewed here. 

## 2. Central vs. Peripheral Axonal Regeneration

Growth of developing axons requires a process of polarized axonal extension, which stops once axons reach their targets and establish new connections. This growth ability at embryonic stages is progressively lost in adulthood, but can be revived by peripheral neurons upon injury [1,2,3,5]. Indeed, axonal growth also occurs during regeneration of adult peripheral axons, with some differences: regenerating axons need to form growth cones starting from the severed stumps that can be far away from the cell bodies, and often need to cover long distances, with high demand of building materials and transport. 

At variance from PNS neurons, which are able to regenerate to a certain extent, most CNS axons do not re-grow following damage. Indeed spinal cord injuries, traumatic brain injuries, stroke, and various neurological disorders typically result in regeneration failure, with poor prognosis for patients. The differences in regeneration capabilities between peripheral and central neurons rely on both intrinsic qualities of the neurons themselves, and on the neuronal environment, mainly glial-driven, which impacts the regenerative outcome [1,2,3,4,5,6,17,18,19]. The finding that some injured CNS neurons are able to re-grow into grafted permissive substrates has paved the way for the identification of inhibitory factors associated with glial scars, myelin debris, etc. [18], that account for the inhibitory influence of CNS glia on regeneration of central neurons.

The preconditioning lesion of primary sensory neurons from dorsal root ganglia (DRG) has been extensively explored to study mechanisms regulating axonal regeneration. As adult DRG neurons are competent for regeneration, the conditioning lesion paradigm has provided an unique model to investigate how the regenerative program is turned on or off. DRG neurons are pseudobipolar neurons with only one axon originating from the cell body. However, the axon branches out to two axons: the peripheral branch innervates the sensory organs and receives sensory information from the periphery, while the central branch enters the spinal cord to relay the sensory information. The two axonal branches from the same cell body are fundamentally different in their responses to injury, and this is largely due to their environments: indeed, while the peripheral branch readily regenerates upon injury, the central branch cannot. However, a first injury at the peripheral branch, called a conditioning lesion, can increase the regenerative responses to a second lesion occurring at either the peripheral or the central branches [20,21]. Moreover, injured central axons from DRG neurons with a conditioning lesion can partially regenerate across the hostile spinal cord injury sites [22]. These studies have provided insights into how ‘sleeping’ intrinsic growth activity can be reactivated in adult neurons, i.e., by down-regulation of genes encoding for neuronal activity (ion channels, synaptic proteins) [23], and up-regulation of growth-associated genes and related transcription factors, the so-called RAGs (regeneration-associated genes) [4], normally repressed by target-derived signals [24,25]. This has paved the way for manipulation of intrinsic programs to force nerve regeneration [25,26]. 

Beside neuron-intrinsic gene programs, several extrinsic factors contribute to regeneration failure/competence. SC, the glia of the PNS, ensheath and myelinate peripheral axons and secrete a basal lamina (BL), composed of growth-promoting laminin, type IV collagen, and heparin sulfate proteoglycans. The abundance of the BL, together with the up-regulation of extracellular matrix (ECM) proteins by SC upon injury, support nerve regeneration in the PNS [27]. Conversely, oligodendrocytes in the CNS do not secrete any BL. Moreover, while the major obstacles for axon regeneration in the CNS are myelin-associated inhibitory molecules and glial scars produced by astrocytes after injury, in the PNS, SC and macrophages rapidly remove myelin debris upon damage, and SC downregulate myelin proteins. Thus, the different local reaction after injury is an important factor that contributes to the ability of the PNS to regenerate [1,27].

Beside the different behaviour of PNS and CNS glial cells in response to injury, additional factors contribute to the regenerative outcome, i.e., microtubule stabilization of axons and growth cones (occurring in the PNS, not in the CNS), which, by preventing the formation of retraction bulbs, allows growth cone formation [1,3,17,28,29]. Indeed in mammals, upon a crush injury to the sciatic nerve, motile growth-cone like structures form within hours, while severed CNS axons form swollen and disorganized dystrophic endings [30,31]. In support of these observations, microtubule stabilization promotes CNS regeneration and functional recovery after injury [28]. One additional mechanism that may limit regeneration in the CNS is the lack/inadequacy of local protein synthesis in central axons, an emerging topic requiring further investigation. Epigenetic mechanisms are also emerging important contributors of nerve regeneration (see below). 

## 3. The Injury Response in the PNS: Intrinsic and Extrinsic Controls of Axonal Regeneration

### 3.1. Local Protein Synthesis in the Axon

Upon peripheral nerve injury, a fast, massive calcium influx occurs in the axoplasm [32,33], leading to a depolarization wave along the axon to the neuronal soma, influencing gene expression [34,35]. Beside this back signaling, local calcium accumulation at the injury site triggers calpain-dependent structural rearrangements of the cytoskeleton that facilitate membrane resealing [36], which relies on calcium-regulated proteins [37], and on growth cone formation [33,38,39]. Moreover, local calcium increase it engages MAPK retrograde signaling through phosphorylation by DLK, which in turn contributes to gene expression activation [40]. This ‘fast’ signaling is followed within hours by a ‘slow’ one, dynein-mediated, which relies on retrograde, motor-based axonal transport [35,41]. Translational machinery and mRNAs are transported from the cell soma to the damage site [42], but extra-neuronal sources have also been reported (e.g., a direct transfer of ribosomes from SC to the axon) [43]. Local translation in the axon is a key component of the injury response mechanism, and it is crucial for the regenerative outcome: on one hand it provides new molecules to sustain axonal re-growth [44], on the other it generates signals that are sent back to the cell soma to engage pro-regenerative and pro-survival pathways [35,45]. Indeed, upon a peripheral lesion, mTOR (the mammalian target of rapamycin), whose expression increases at the injury site [46], acts as a core regulator of the process in the axon, promoting translation of its own mRNA, as well as of additional transcripts, among them those encoding for importin β1, vimentin, RanBP1, STAT3, and PPARPγ [46]. Furthermore, perturbation of the mTOR pathway affects nerve re-growth in the PNS [46]. 

### 3.2. Epigenetic Regulation of Axonal Re-Growth Program

It is now clear that the cellular phenotype is determined by DNA and chromatin modifications. Indeed, epigenetic changes represent the phenotypic adaptation of the organism as a result of the interaction with the surroundings. The coordination among genetic and epigenetic programs is crucial for both development and regeneration [4,47,48]. During development, once axons have reached their targets, several transcriptional pathways regulating axonal growth are turned off. Upon injury, peripheral neurons, at variance from central ones, are able to successfully initiate an orchestrated transcriptional response [4]. To do this, the cell soma must receive signals carrying information about the status of the periphery via retrograde trafficking. These alert or alarm signals, which may be represented by mediators synthetized in the axons, by the lack of trophic factors, or by calcium waves propagating along the axon, activate a regenerative genetic program, with the contribution of epigenetic mechanisms, e.g., acetylation and methylation of histone proteins, DNA methylation, and microRNAs affecting the expression of RAGs (reviewed in [48,49]). A decline in the ability to engage such a transcriptional activation occurs during aging [50], mainly because of a reduced chromatin accessibility at gene regulatory regions, and a decrease in the levels of transcription factors. 

As chromatin accessibility to transcription factors and regulatory elements is crucial to allow gene transcription (e.g., a more relaxed configuration favours the process), epigenetic mechanisms affecting chromatin structure are likely to play a role in nerve regeneration. Histone acetylation, methylation, phosphorylation, and other epigenetic modifications affect the chromatin state, activating or repressing gene transcription. The level of histone acetylation is primarily regulated by the balance between the activities of histone acetyl transferases (HATs) and histone deacetylases (HDACs). HDACs, which work to maintain a closed chromatin structure, represent important components of the epigenetic pathway that regulates regeneration by counteracting HAT activity [27,51,52]. Injury-induced calcium waves induce the PKCγ-dependent nuclear export of HDAC5, enhancing histone acetylation and gene transcription [53]. More recently, HDAC3 signaling has been recognized as a brake to axonal regeneration: indeed HDAC3 inhibition overcomes the inability of sensory axons to regenerate upon a spinal cord injury [54]. This evidence led to the hypothesis that comparable regulatory mechanisms activating epigenetic regenerative responses may be not functional in the CNS, preventing the gene expression changes essential for efficient axonal re-growth. 

Injury-induced ERK engagement at the cell periphery, and its retro-transport to the cell soma, contribute to epigenetic changes supporting nerve regeneration by activating PCAF, a HAT, by phosphorylation [55]. Thus, epigenetic mechanisms are key factors that initiate and sustain the regenerative genetic program, and epigenetic regulators represent a group of potential drug candidates to modify the transcriptome, thus promoting efficient axonal regeneration [56,57].

Also, non-coding RNA transcripts (miRNAs, siRNAs, lncRNAs) regulate gene transcription via epigenetic mechanisms. Their relevance in nerve regeneration is a promising, open field of investigation [58].

### 3.3. The Importance of a Pro-Regenerative Environment: Schwann Cells as a Central Hub Coordinating Multiple Partners 

Peripheral nerve regeneration relies on intense and coordinated communication between the severed axon and the environment [59,60]. Successful nerve re-growth and re-establishment of functional contacts are achieved through an orchestrated interplay between different cell types in the milieu. SC, the glia of the PNS, orchestrate peripheral nerve regeneration working as a central hub, collecting signals from neurons, fibrobasts, endothelial and inflammatory cells, and undergoing a complex transdifferentiation program to sustain nerve re-growth (Figure 1). Given the lack of stem/progenitor cells in adult peripheral nerves, regeneration is sustained mainly by resident differentiated cells, whose plasticity is crucial for nerve repair [16,61,62]. 

Seminal studies have led to the identification of crucial molecules and pathways driving peripheral nerve regeneration, mainly employing the compression (crush) or transection (cut) of the sciatic nerve as experimental models of nerve injury. After nerve crush, BL tubes remain intact along the original nerve length, allowing axons to re-grow following their original trajectories, thus achieving a more complete functional recovery. At variance from the crush, nerve transection cuts both axons and BL. Continuity between the proximal stump and the target is lost and, despite the formation of a new tissue called bridge, reconnecting the two stumps, many axons are likely not to find the appropriate BL tubes in the distal stump to reach their natural targets, making regeneration incomplete [5]. Indeed, in clinical practice, surgeons try to re-attach the two nerve stumps by apposing the two cut nerve ends to allow rapid entry of axons into the distal stump and maximum re-growth through it to the target tissue. They also attempt to align fascicles within the two nerve stumps to help axons find their original BL tubes (reviewed in [63]). 

The cut and crush experimental models of traumatic injuries led to the realization that developmental and regenerative programs in SC overlap at least partially, but are distinctly different. Moreover, they highlighted that: (i) no matter whether damage is caused by crush or cut, the response of SC of the distal stump (distal SC) is the same; (ii) distal SC and ‘bridge’ SC undergo a different transcriptional reprogramming, reflecting the influence of different microenvironments [64].

#### 3.3.1. The Injury Response of Distal SC 

Following nerve transection, the distal part of the axon degenerates. Functional recovery requires that the proximal segment of the axon elongates back to its target through a bridge that forms at the injury site. This is achieved thanks to a remarkable transcriptional reprogramming of distal SC that do not merely dedifferentiate, but rather establish a novel differentiated state, and are often referred to as Repair Schwann cells (Repair SC). Repair SC consist of a transient cell population, which exists only when needed. Their transcriptional reprogramming is functional to cell proliferation, clearance of nerve debris and myelin (myelinophagy by SC aided by recruited macrophages) [65], and to the morphological changes required to allow their elongation along the BL to guide the axon. Indeed, these cells adopt an elongated shape with fine processes, they are often branched, and are 2–3 times longer than the myelin and Remak SC found in uninjured adult mouse nerves [66]. Distal SC attract inflammatory cells through cytokine release (e.g., Il-1β, Il-1α, TNFα, LIF), which participate in the regenerative process, and themselves secrete several neurotrophins (GDNF, artemin, BDNF, NGF) [67,68], essential for the survival and the re-growth of severed axons. Macrophage response is controlled and modified by SC and fibroblasts through secreted factors [60]. Once the repair process is completed, SC exit the cell cycle and re-differentiate, and inflammation is resolved, although recovery may be not fully achieved, depending on the extent of injury, and of ECM accumulation.

SC ‘transdifferentiation’ is a complex phenomenon, involving the down-regulation of myelin-associated genes, the revival of developmental pathways, together with the engagement of unique gene signatures (reviewed in [16]). Following nerve injury, SC respond to axonal damage with a strong, sustained activation of the ERK signaling pathway, both at the injury site and throughout the distal stump [69]. Sustained Raf/MEK/ERK signaling in the absence of neuronal damage is sufficient to drive SC to switch to a dedifferentiated state, supporting a central role for ERK signaling in SC in orchestrating nerve repair [70], a pathway also required during development and for myelination. Another signaling pathway working in SC both during development and in regeneration is the one initiated by the engagement of Gpr126, a member of the GPCR (G-protein couled receptor) family: by interacting with laminin in the BL, Gpr126 acts by elevating the cAMP levels within SC. Without Gpr126 in SC, axonal regeneration and reinnervation are impaired, and remyelination is absent [71]. 

A master regulator of the unique properties of Repair SC is the transcription factor c-jun [67,68], while the same molecule is not essential for SC development. c-jun orchestrates the repair program by controlling the expression of 172 genes, among them Sonic Hedgehog and Olig1 (reviewed in [16]), not involved in SC development. Again, differently from developing SC, Repair SC express high levels of cytokines through which they recruit macrophages (reviewed in [72,73]), exploit autophagy to clear myelin [65], and adopt an elongating morphology required to work as a guide for the re-growing axons [66]. Moreover, they display chromatin modifications, which adapt them to the regenerative requirements (reviewed in [48,74,75]). Among the different epigenomic changes, DNA methylation has turned out to be of particular importance for Repair SC phenotype, acting as a repressive modality of gene expression [75,76].

#### 3.3.2. Bridge Formation

Following nerve transection, a new tissue called bridge forms to reconnect the 2 axonal stumps. It consists mainly of macrophages, endothelial cells, SC, and fibroblasts. Without the correct orientation, the re-growing axon would hardly find its way through the bridge, with few chances to reconnect with the original target. Successful nerve bridge assembly, and proper migration of both SC (which work as tracks to guide the axon) and the axon are the result of the cooperation of many cell types and mediators at the injury site (Figure 1). Monocyte-derived macrophages secrete VEGF in response to hypoxia, to induce a polarized vasculature that is used by SC as a track to carry re-growing axons across the bridge [77]. Beside VEGF, macrophages also secrete the glycoprotein Slit3 which, by interacting with Robo1 in SC, keeps migrating SC in the bridge [78]. Slit3, together with other classical guidance molecules (netrins, etrins, ephrins, etc.), participates in the communication between SC, nerve, macrophages, and fibroblasts at the injury site, and controls the correct bridge formation and axonal pathfinding required for proper regeneration [78,79,80]. Moreover, Netrin1/DCC signaling between migrating SC (releasing Netrin1) and regenerating axons (expressing the Netrin1 receptor DCC) directs the re-growing axons across the bridge [81,82]. 

Increasing evidence indicates that the cross talk between SC and nerve fibroblasts is crucial for successful regeneration across the bridge: indeed, upon sciatic nerve cut, fibroblasts in the bridge area release EphrinB2 which, by binding to the EphB2 receptor expressed on SC, induces the SOX2-dependent translocation of N-cadherin on the SC surface [83] (together with Robo1 expression [78]), which in turn promotes collective cell migration. Collective cell migration, frequent in development, generates a more efficient directed migration than single cell migration, likely providing a more continuous substrate for re-growing axons. By promoting cell adhesion, this pathway directs the formation of SC cords within the bridge, fundamental for the correct elongation of regenerating axons [83]. Nerve fibroblasts, accumulated at the lesion site, abundantly express Tenascin C, a glycoprotein associated with the ECM of vertebrates, which stimulates SC migration through a β1-integrin-mediated pathway [84]. Fibroblasts at the injury site also secrete TGFβ. By synergizing with Ephrin signaling, TGFβ promotes SC collective migration (N-cadherin and SOX2 dependent) from the proximal stump across the bridge to reconnect with the outgrowth into the bridge from the distal stump. Beside this action, TGFβ is a well-known trigger of the EMT (epithelial to mesenchymal transition) [64]. EMT is a cellular reprogramming process characterized by the down-regulation of molecules that promote cell adhesion and provides increased cell motility, plasticity, migration, and proliferation that help injury-induced tissue remodeling. Mainly studied in development and cancer, it is now established that this process is a normal physiological response to injury [61]. As a consequence of a localized TGFβ release at the damage site, SC of the bridge (where arguably tissue remodeling is more radical than in the distal site) show an evident EMT-signature (with enhanced proliferation rate and mesenchymal phenotype) [64], while SC within the distal stump show a less marked but measurable EMT response [61,76]. This means that the microenvironment of the nerve bridge imposes a distinct phenotype from de-differentiated SCs to ensure successful reinnervation. Thus, extrinsic signals can directly modulate intrinsic programs to adapt local cell function to the specific requirements of the surrounding tissue. TGFβ localized signaling is responsible for the regional reprogramming of the bridge, which likely explains why SC show different properties depending on their location along the nerve with respect to the lesion site (i.e., distal SC undergo an injury-induced transcriptional program different from that of bridge SC [64]), thus adapting themselves to meet the particular needs that arise after injury. With the transcriptional profile of bridge SC being reminiscent of partially dedifferentiated cancer cells (e.g., they increase the expression of mesenchymal and Myc targets and proliferate more that distal SC), the identification of signals and pathways driving nerve repair may also be relevant to cancer research, and could help to identify specific target molecules to counteract cancer progression.

### 3.4. The Mechanobiology of Schwann Cells 

Cells can sense environmental changes through mechanosensors, which relay information about the stiffness of the milieu, whose variations are converted into intracellular signals via mechanotransducers. Cells are constantly subjected to tension forces mediated by the cytoskeleton and to compression by the ECM and neighbours. Such mechanical forces are responsible for biochemical and transcriptional changes, which have just started to be identified, through which they influence important cellular processes such as migration, proliferation, and differentiation.

During development, the SC myelinating fate is determined by the amount of type III neuregulin-1 expressed on axonal membranes [85], and by the physical contact with the axon [86], which directs BL deposition by SC. Thus, interaction with both axons and the ECM is fundamental for SC fate and occurs via mechanosensors such as adhesion complexes and the cortical cytoskeleton [87]. Then, these stimuli are transduced into biological responses by several pathways, among them those that involve the transcriptional activators YAP/TAZ [88,89]. Mechanotransduction via YAP/TAZ is crucial during development, as it regulates SC proliferation, differentiation, and myelination [88,90], while YAP/TAZ their role after myelination is contradictory [90,91]. Recently, it has been shown that they are required for myelination, but not for Repair SC transdifferentiation and proliferation after nerve injury [92]. 

Most of the present knowledge on mechanobiology derives from in vitro studies, based on manipulation of the stiffness of the cellular milieu, while in vivo manipulation of mechano-pathways is much more challenging. As several different signaling pathways converge in YAP/TAZ engagement, their activation cannot be directly ascribed to a mechanical stimulation.

Unlike YAP/TAZ, the engagement of mechanosensitive ion channels, e.g., potassium channels and Piezo ion channels, can be irrefutably inconfutably attributed to mechanostimulation [87]. These channels work as mechanosensors as well as mechanotransducers by sensing membrane changes and converting them into electric or biochemical signals. In Drosophila, the Ca^2+^ permeable non-selective cation channel Piezo has been recently described as an inhibitor of axonal regeneration via the CamKIINos-PKG pathway [93].

### 3.5. Presynaptic Neurotoxins: Valuable Tools to Identify Alarm Molecules and Signaling Pathways Driving Motor Axon Regeneration 

While the number of intracellular signaling systems regulating SC plasticity is rapidly increasing, much less is known about the nature of the extracellular signals, generated in response to nerve injury, responsible for their engagement. As the injury response by SC starts very early after damage, well before axonal degeneration and macrophage recruitment, axons are likely the major source of alarm molecules. 

The molecular mechanisms underpinning successful nerve regeneration have been widely investigated in different animal and injury models. Many aquatic vertebrates display remarkable abilities to regenerate limbs and tails after amputation, and these properties make them ideal model organisms to identify mediators and signaling pathways underpinning the regenerative response to injury [94]. Moreover, the growth-permissive environment of their CNS has been useful both for the identification of factors differing between mammals and fishes that may account for differences in CNS regeneration, and for the characterization of conserved intrinsic pathways that regulate axon regeneration in all vertebrates [95].

The NMJ, the specialized synapse controlling locomotion, is a privileged site of study of the cross talk driving nerve repair [96]. It is the site where the electric signal running along the motor axon is converted into a chemical one in the form of a released neurotransmitter (acetylcholine, ACh), which crosses the synaptic cleft and binds to muscle nicotinic ACh receptors, triggering muscle contraction [7]. The NMJ is formed by the motor axon terminal and the muscle, separated by a BL, and covered by PSC. This synapse is exposed to many injuries including traumas and animal and bacterial toxins. In addition, it is a primary site of pathogenesis in many neurological conditions including myasthenia gravis and related disorders, congenital myasthenic syndromes, Guillain-Barrè syndromes, and motor neuron diseases such as amyotrophic lateral sclerosis and spinal muscular atrophy [97,98,99,100]. Remarkably, the NMJ has retained throughout evolution a striking capability to regenerate. Hence, the definition of the molecular interplay between NMJ components during nerve terminal injury, and more importantly during NMJ reorganization/plasticity and the ensuing regeneration, may lead to the discovery of potential candidates to stimulate regeneration even in central neurons. 

In rodents, the process of degeneration/regeneration of the NMJ has been mainly studied following the compression or transection of the sciatic nerve. These traumatic injuries cause a process called Wallerian degeneration [73,101], characterized by a strong inflammatory response which, while helping nerve regeneration, makes the identification of signaling molecules responsible for the process challenging. Recently, the use of presynaptic neurotoxins to induce a reversible damage restricted to the motor axon terminal, with no inflammation, has turned out to be a powerful tool to identify signals and pathways driving nerve plasticity and regeneration [96,102,103]. 

α-latrotoxin (α-LTx), a spider pore-forming toxin, specifically attacks the presynaptic membrane inducing its rapid but reversible degeneration through a massive calcium overload [104,105,106,107,108,109,110]. In mice, the entire degeneration and regeneration process induced by α-LTx takes a few days, with complete anatomical and functional recovery of the synapse [102,106]. Degenerating nerve terminals release mitochondrial alarm signals (i.e., mitochondrial DNA and cytochrome c), which activate the ERK1/2 pathway in SC in vitro and in vivo [102], and ATP, thus contributing to SC activation through Ca^2+^ and cAMP pathways [111]. A major player in NMJ recovery of function is hydrogen peroxide (H_2_O_2_), rapidly generated by mitochondria of injured neurons as a consequence of calcium overload. H_2_O_2_ drives neurotransmission rescue via ERK1/2 signaling [102] (Figure 2).

#### 3.5.1. Hydrogen Peroxide: A Key Alarm Signal Driving Peripheral Nerve Regeneration

Reactive oxygen species (ROS) have long been considered deleterious molecules damaging cellular integrity and function. It is now becoming increasingly evident that ROS also contribute to many important physiological processes, such as tissue healing and regeneration [112,113,114,115]. H_2_O_2_, the most stable among ROS species and easily diffusible, holds the best features of an intercellular mediator [116]. It is generated in all aerobic organisms as a byproduct of physiological cellular processes and, in order to prevent its toxic effects, the same organisms are equipped with detoxifying enzymes such as catalase, glutathione peroxidases, and peroxiredoxins [117,118]. All mammalian cell types produce H_2_O_2_ as a signaling molecule in response to a variety of extracellular stimuli. Its production can be regulated spatially (by compartmentalization of H_2_O_2_ sources and targets) and temporally (by inactivating enzymes) [119].

In zebrafish, H_2_O_2_ levels are spatially and temporally regulated during early development, reaching high levels during morphogenesis, which progressively decline in mature tissues. A reduction in H_2_O_2_ levels impairs axonal projections by retinal ganglion cells, and this phenotype can be rescued by activation of the Hedgehog pathway [120]. Also during adult regeneration, axonal growth is controlled by the interplay between H_2_O_2_ and Hedgehog [121,122]. Experimental evidence in different animal models shows that a rapid concentration gradient of H_2_O_2_ is generated upon injury [112,123], and that H_2_O_2_ is a powerful chemoattractant for leukocytes [112,124,125]. Importantly, lowering ROS levels by pharmacologic or genetic approaches impairs regeneration [112,123].

The identity of H_2_O_2_ signaling targets mediating axonal regeneration remains an open question. H_2_O_2_ modulates signaling by oxidation of thiol groups of cysteines and methionines in target proteins, leading to structural and functional modifications [115]. In addition, H_2_O_2_ inactivates the phosphatase PP2A, resulting in sustained activation of MAPKs (i.e., ERK1/2) [126], and of other proteins involved in nerve regeneration, such as members of the Src family [127,128]. Cytoskeleton proteins, whose dynamic assembly is regulated by ROS signalling [129], are also attractive candidates. Releasing brakes (for example, by inhibition of Rho GTPases by oxidation) is among the mechanisms that promote axonal re-growth [130]. 

H_2_O_2_ is able to rewire in primary SC the expression of several RNAs at both transcriptional and translational levels, among them genes involved in cytoskeleton remodeling and cell migration, with the Annexin (Anxa) proteins being the most represented family [131]. Both Anxa2 transcripts and proteins accumulate at the tip of long processes that SC extend upon H_2_O_2_ exposure and, interestingly, SC reply to this signal by locally translating Anxa2 in pseudopods, and by undergoing extensive cytoskeleton remodeling [131]. Hence, similar to neurons, SC take advantage of local protein synthesis to change shape and move toward damaged axon terminals to facilitate axonal regeneration in an H_2_O_2_-dependent manner. These observations support the view that redox signaling via paracrine communication, together with calcium signaling, are ‘immediate injury signals” that integrate early damage response with late regeneration [128,132]. 

H_2_O_2_, detected by specific probes [133], is rapidly produced by neuronal mitochondria upon injury by α-LTx [102], or by an anti-ganglioside plus complement complex (which models several aspects of the Miller Fisher syndrome) [134], as a consequence of calcium overload, and appears to be a major player in nerve repair by engaging the ERK1/2 pathway in PSC. An increase in local calcium concentration, toward which mitochondria move to buffer excessive calcium ions, is a fast event shared by many peripheral nerve injuries. Calcium overload within mitochondria, enriched in the axon, is a known trigger of mitochondria dysfunction and ROS (H_2_O_2_) production [135,136]. 

Along this line it was recently reported an unconventional mechanism by which inflammation associated with sciatic nerve injury induces ROS production in motor neuron cell bodies, which in turn supports the regenerative program. Macrophages recruited at the lesion site release exosomes containing functional NADPH oxidase 2 (NOX2) complexes that, once endocytosed by axons, are retrotransported in a dynein-dependent fashion to the cell body. Here, macrophage-derived NOX2, by oxidizing PTEN, stimulates the PI3K-AKT signaling pathway, in turn promoting regeneration [137].

#### 3.5.2. Reactivation of the Developmental Axis CXCL12α-CXCR4 Promotes Peripheral Nerve Regeneration

G protein-coupled receptors are a large protein family that detects molecules outside the cell and activates intracellular signal transduction pathways through G protein-coupled receptor kinases [138,139]. A member of this family, named CXCR4, and its natural ligand, the chemokine CXCL12α, have been recently described as important players in the functional and anatomical recovery of the NMJ following acute damage to presynaptic nerve terminals [103]. CXCL12α was initially discovered by screening for proteins carrying a signal sequence either for secretion or for incorporation into the plasma membrane [140], and by a search for growth factors for pre-immune cells. Indeed CXCL12α, also termed stromal cell-derived factor 1 (SDF-1), is a growth factor for bone marrow pre-B cells [141], and it plays a variety of additional roles in the immune system. Later on, CXCL12α was shown to be involved in the development of various regions of the CNS [142,143,144,145,146,147,148,149,150,151]. In response to a motor axon terminal injury, PSC synthesize and release CXCL12α which, by interacting with CXCR4 expressed on the tip of the re-growing axon, promotes muscle re-innervation [103] (Figure 2). This represents an additional, remarkable example of the revival of the developmental, pathfinding signaling axis, CXCL12α-CXCR4 [147], during nerve regeneration. Interfering with either the two members of the axis delays regeneration [103], while exogenous stimulation of CXCR4 via a receptor agonist remarkably speeds up neurotransmission rescue following an acute injury to the presynaptic nerve terminal, or the crush/cut of the sciatic nerve, advocating the CXCR4 receptor as a pharmacological target to promote nerve regrowth [152,153].

### 3.6. Regeneration at Sensory Terminals

Differently from the regeneration of motor axon terminals, very little is currently know about the same process occurring at sensory endings. Although the most common experimental approach to study peripheral nerve regeneration, i.e., a traumatic lesion to the sciatic nerve, affects both sensory and motor axons, successful regeneration is usually ascribed to the attainment of a complete recovery of motor functionality; the latter can be assessed by electrophysiological read-outs, which allow a quantitative and objective evaluation of the extent of regeneration, differently at variance from sensory tests that are highly influenced by the biological variability on one hand, and by the motor function condition on the other. Moreover, the availability of tools such as neurotoxins that selectively and reversibly damage the motor axon terminals has provided additional molecular information on the regenerative process at these terminals. 

Most of the present knowledge on the regeneration of sensory neurons comes from studies on DRG neurons. Almost all somatosensory pathways begin with the activation of these cells, which are responsible for thermoception, nociception, mechanoception, and proprioception [154]. Following transection of the peripheral axon, adult sensory neurons undergo a series of degenerative/regenerative events well described for transected motor axons [59]. Local protein synthesis in regenerating axons has also been reported [155]. Beside an inherent capacity of DRG neurons for regeneration, the surrounding tissue impacts the degree of recovery after injury. Indeed, it has been reported that SC express distinct sensory and motor phenotypes and thus support regeneration in a phenotype-specific manner [156]. Additional contributors to the regeneration of sensory neurons are satellite glial cells (SGC), which completely envelop the neuronal soma: these cells allow close communication and molecular interchange and undergo injury-induced transcriptional changes, thereby forming a functional unit with the sensory neuron to orchestrate nerve repair [154,157]. 

Among the PNS nerves, the olfactory nerve is particularly adept at regeneration, as it continuously regenerates throughout life. It contains peculiar glial cells named OEC (olfactory ensheathing cells), at variance from most of the PNS nerves that are populated by SC. OEC and SC cells share many common features, but display different properties during regeneration after injury, which have been widely described, reflecting different environmental requirements [158]. 

A previously unrecognized specialized cutaneous glial cell type, called nociceptor SC, has been recently described [159]. In direct functional connection to unmyelinated nociceptive fibers, these cells are inherently mechanosensitive and transmit nociceptive information to the nerve. Beside their potential role in chronic pain, it will be important to investigate how these nociceptive SC respond to axonal injury and disease states.

An interesting, and amenable to manipulation, experimental system to evaluate both motor and sensory axonal regeneration is represented by muscle spindles (intrafusal muscle fibers) that lie parallel to skeletal (extrafusal) muscle fibers. While extrafusal muscle fibers generate force via muscle contraction to initiate skeletal movement, muscle spindles are proprioceptive receptors in the skeletal muscle that respond to the length of the muscle, thus playing an important role in motor control. Muscle spindle sensory output is sent to the CNS, where it is utilized to regulate motor activity through α-motor neurons signals to extrafusal skeletal muscle fibers, and to regulate proprioreceptor sensitivity through γ-motor neurons signals to intrafusal fibers [160]. Following peripheral nerve injury, structural changes occur in muscle spindles, which undergo atrophy [161]; given that a successful motor function recovery relies also on effective regeneration of this system, muscle spindles could be a promising experimental model to compare the regenerative capabilities of sensory and motor terminals, thus helping to fill the gap in knowledge on regeneration at sensory endings.

## 4. Conclusions

Peripheral nerves regenerate thanks to intrinsic and extrinsic contributions. In response to nerve damage, both neurons and SC convert to cell states specialized to deal with injury and to promote repair. PNS neurons respond to an insult by activating a pro-regenerative genetic reprogramming, under epigenetic regulation, and engage local protein synthesis in the axon. In addition, complex and coordinated multi-cellular cross talk orchestrates the injury response. SC play a central role in the process, working as a hub to coordinate multiple signals and responses. They undergo a transcriptional reprogramming involving the reviving of developmental pathways on one hand, and the engagement of new and dedicated processes on the other. Following nerve transection, a heterogenous population of SC can be detected, whose behaviour depends on their anatomical localization with respect to the lesion site, and on the localized signaling generated by the environment, in order to adapt to the particular needs that arise upon damage. Hence, neuronal degeneration, via delivery of alarm signals, triggers SC transdifferentiation through activation of cell-intrinsic transcriptional programs. Then, extrinsic signals from the microenvironment superimpose on these programs in a context-dependent manner to adapt SC function to the specific repair requirements of the surrounding tissue. Intriguingly, the transcriptional profile of bridge SC is reminiscent of that of cancer cells, and most solid tumors are very similar to wounds. Thus, the identification of signals and pathways driving wound repair is relevant also to cancer research and could help to identify specific candidate molecules to block cancer.

SC ‘sense’ environmental modifications also through mechanosensors and mechanotransducers, which relay information about changes in the stiffness of the milieu, and whose identification is ongoing. The mechanobiology and mechanosensing of SC is an emerging and relevant topic in the regeneration field: indeed, it is currently unknown how compression, demyelination or loss of axons affects the stiffness of peripheral nerves, thus in turn influencing SC pro-regenerative behaviour. Understanding how mechanical properties affect the cellular and molecular biology of SC during development, myelination, and following injuries will open new insights into the regulation of PNS development, as well as in treatment approaches for peripheral neuropathies.

The discovery of mediators and pathways driving the process of peripheral nerve repair holds great translational relevance, as it provides new promising candidates and approaches to promote nerve recovery of function after different forms of peripheral nerve injury, which could be tested to force regeneration also in the CNS. In this respect, the NMJ turns out to be a privileged point of observation, whose injury response provides a valuable source of mediators with pro-regenerative potential to be tested in different forms of nerve damage. 

Very little is currently know about the regeneration of sensory endings. We propose muscle spindles as an useful experimental model system to gain insights into the process.

## Figures and Tables

**Figure 1 cells-09-01768-f001:**
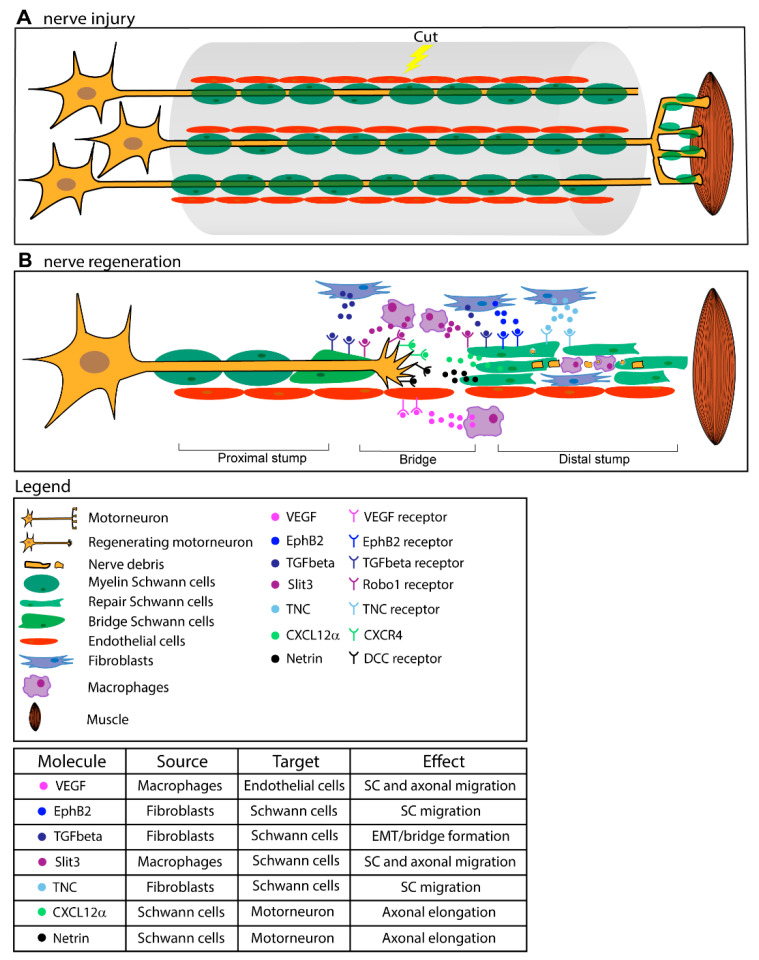
Schematic overview of the orchestrated response to a peripheral nerve injury. Nerve cut generates a gap between the two cut ends, and the distal one progressively degenerates. Repair Schwann cells (SC) (green) in the distal stump, activated by signals coming from the degenerating axon (yellow), are responsible for the clearance of nerve and myelin debris, the recruitment of macrophages (purple), and the secretion of neurotrophic factors. Within a few days, a bridge forms to reconnect the two stumps. Signals from endothelial cells (red), fibroblasts (blue), and macrophages guide collective and oriented SC migration through the bridge, allowing proper axonal re-growth.

**Figure 2 cells-09-01768-f002:**
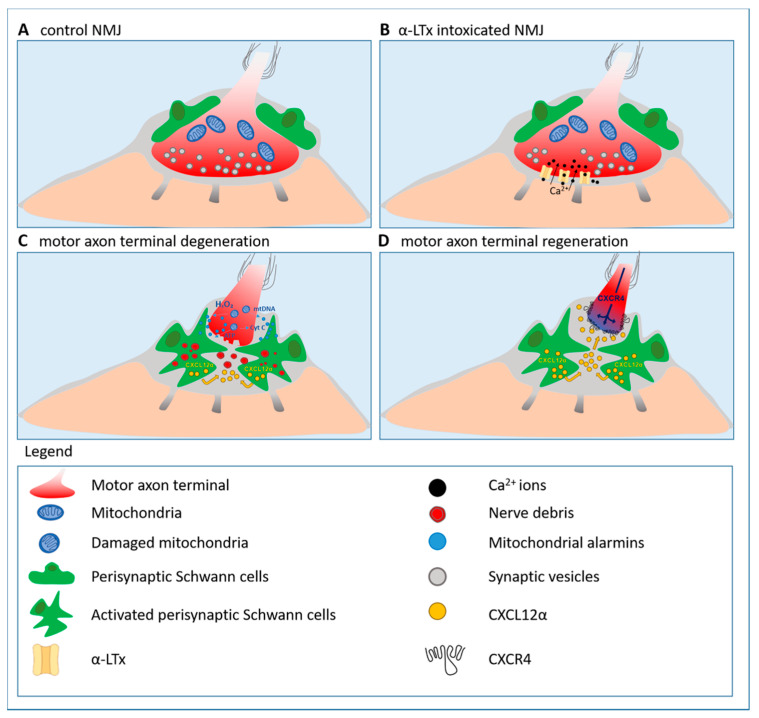
Motor axon terminal regeneration at the murine neuromuscular junction (NMJ) following the acute and reversible injury by the pore-forming toxin α-Latrotoxin (α-LTx). Neuronal alarm signals, mainly of mitochondrial origin, such as H_2_O_2_, cytochrome c, mitochondrial DNA (mtDNA), and ATP (blue spots), trigger perisynaptic Schwann cells (PSC) (green) activation. Activated PSC phagocytose nerve debris (red), and release the chemokine CXCL12α (yellow) which, by interacting with CXCR4 re-expressed by the motor axon stump, promotes motor axon elongation and, in turn, NMJ functional restoration (adapted from [103]).

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
