# Peer review of "Signals Orchestrating Peripheral Nerve Repair"

_cells, 2020, doi:10.3390/cells9081768_

Round 1

Reviewer 1 Report

This is an exciting and impressive review in which the authors systematically review the main intrinsic and extrinsic mechanisms that contribute to peripheral nerve regeneration. They also highlight the important role of Schwann cells as central hubs in this process. In summary, I think this is an exciting and novel contribution that will be of broad interest to the field and will be highly cited. I do not have any major concerns about this review, but the manuscript could be improved by addressing the following issues:

Minor Points

  1. For the figures, please label different panels with “A”, “B”, and “C”. It is much easier to understand the figures after labeling. Please do not capitalize the headings.
  2. In figure 2, the legends are small and difficult to see. Please use different colors to make the contrast more obvious.

Grammatical and typographical errors:

  1. Line 53. Should read: “Indeed, …”
  2. Line 108. Should read: “Upon peripheral nerve injury, a fast…”
  3. Line 145. Should read: “phosphorylation, and other epigenetic modifications”
  4. Line 256. “well known” should be “well-known”
  5. Line 288. Should read: “During development, SC myelinating fate”
  6. Line 305. Should read: “In Drosophila,”
  7. Line 336. Should read: “In rodents, the process of…”
  8. Line 376. Should read: “In zebrafish, H2O2 levels…”

Author Response

Reply to Reviewer 1.

We thank the Reviewer for his/her appreciation for the review.

  1. For the figures, please label different panels with “A”, “B”, and “C”. It is much easier to understand the figures after labeling. Please do not capitalize the headings. Amended in the revised figures.
  2. In figure 2, the legends are small and difficult to see. Please use different colors to make the contrast more obvious. Amended in the revised figures.
  3. Grammatical and typographical errors have been amended in the revised manuscript (in red).

Reviewer 2 Report

In the manuscript “Signals Orchestrating Peripheral Nerve Repair” Rigoni and Negro review the current state of the art in peripheral nerve de- and regeneration, with a particular focus on novel signals discovered at the NMJ. The authors first briefly introduce the involved cell types along the peripheral nerve and of the NMJ. They then review the distinct settings in the PNS (permissive) and the CNS (restrictive to regeneration). The main part of the manuscript covers intrinsic regenerative programs in the axon and the Schwann cells, featuring an in-depth description of the Schwann-cell guided repair after nerve crush/cut. Finally, the authors elaborate on recently discovered ROS and chemokine signaling in regeneration of the injured NMJ, thereby highlighting their own work.

This is an excellent review of a (thankfully) still evolving topic, written by experts who actively publish in the field. I recommend it for publication.

I can only suggest to consider some very minor issues with the conclusion part and with spelling/wording:

  • The similarity of bridge Schwann cells (Bungner (repair Schwann) cells as well?) to cancer cells is mentioned in the conclusion, this should be introduced in the main text
  • The authors highlight the potential benefit of results from the regenerating PNS for efforts to force regeneration also in the CNS – this might be mentioned also in the conclusion
  • 1, l.39 “assures” should read “assure”
  • 3, l.117 “trasfer” should read “transfer”
  • 6, l.246 “traslocation” should read “translocation”
  • 263 “mesenchimal” should read “mesenchymal”
  • 7, l.286 “trascriptional” should read “transcriptional”
  • A second time in l.293
  • 11, l.427 better put “roles” instead of “functions”

Author Response

Reply to Reviewer 2.

We thank the Reviewer for his/her appreciation for our review.

The similarity of bridge Schwann cells (Bungner (repair Schwann) cells as well?) to cancer cells is mentioned in the conclusion, this should be introduced in the main text. Amended in the revised manuscript (lines 271-275 page 6, in red).

The authors highlight the potential benefit of results from the regenerating PNS for efforts to force regeneration also in the CNS – this might be mentioned also in the conclusion. Amended in the revised manuscript (lines 471-472 page 11, in red).

  • 1, l.39 “assures” should read “assure”
  • 3, l.117 “trasfer” should read “transfer”
  • 6, l.246 “traslocation” should read “translocation”
  • 263 “mesenchimal” should read “mesenchymal”
  • 7, l.286 “trascriptional” should read “transcriptional”
  • A second time in l.293
  • 11, l.427 better put “roles” instead of “functions”

All corrections have been made in the revised manuscript (in red).

Reviewer 3 Report

The manuscript submitted by Rigoni and Negro make an excellent review of the mechanisms allow peripheral axon regrowth. They notably highlight the role of Schwann cells as major coordinator of signals leading to axon regeneration.  The authors present in an elegant manner, the existing literature, with a focus on neuromuscular junctions. Yet, while in is an excellent review it could be improved by:

  • Developing a paragraph on axon regrowth at the sensory terminals, to contrast the focus on the NMJ and because this is a topic lacking in the field.
  • Reducing the amount of acronyms, which are numerous in several sections of the review.

Author Response

Reply to Reviewer 3.

We thank the Reviewer for his/her appreciation for our review.

  • Developing a paragraph on axon regrowth at the sensory terminals, to contrast the focus on the NMJ and because this is a topic lacking in the field.

Following the Reviewer’s suggestion, we have attempted to address the topic of axonal re-growth at sensory terminals (please see below). However, given the poor knowledge in this field in molecular terms, we wonder whether the informations reported would be adequate to improve the quality of the review. If required by the Reviewer, this part will be added as a separate paragraph to the revised manuscript, and the abstract and the conclusion sections will be modified accordingly.

Regeneration at sensory terminals.

At variance from regeneration at motor axon terminals, very little is currently know about the same process occurring at sensory endings. Although the most common experimental approach to study peripheral nerve regeneration, i.e. a traumatic lesion to the sciatic nerve, affects both sensory and motor axons, successful regeneration is usually ascribed to the attainement of a complete recovery of motor functionality; the latter can be assessed by electrophysiological read-outs, which allow a quantitative and objective evaluation of the extent of regeneration, at variance from sensory tests that are highly influenced by the biological variability on one hand, and by the motor function condition on the other. Moreover, the availability of tools like neurotoxins that selectively and reversibly damage the motor axon terminals has provided additional, molecular information on the regenerative process at these terminals.

Most of the present knowledge on regeneration of sensory neurons comes from studies on DGR neurons. Almost all somatosensory pathways begin with the activation of these cells, which are responsible for thermoception, nociception, mechanoception, and proprioception [154]. Following transection of the peripheral axon, adult sensory neurons undergo a series of degenerative/regenerative events well described for transected motor axons [155]. Local protein synthesis in regenerating axons has also been reported [156]. Beside an inherent capacity of DRG neurons for regeneration, the surrounding tissue impacts on the degree of recovery after injury. Indeed, it has been reported that SC express distinct sensory and motor phenotypes, and thus support regeneration in a phenotype-specific manner [157]. An additional contributor to regeneration of sensory neurons are satellite glial cells (SGC), which completely envelop the neuronal soma: these cells allow close communication and molecular interchange, and undergo injury-induced transcriptional changes, thereby forming a functional unit with the sensory neuron to orchestrate nerve repair [155, 158].

Among the PNS nerves, the olfactory nerve is particularly adept at regeneration, as it continuously regenerates throughout life. It contains a peculiar type of glial cells named OEC (olfactory ensheathing cells), at variance from most of the PNS nerves that are populated by SC. OEC and SC cells share many common features, but display different properties during regeneration after injury, which have been widely described, reflecting different environmental requirements [159].

An interesting and amenable to manipulation experimental system to evaluate both motor and sensory axonal regeneration is represented by muscle spindles (intrafusal muscle fibers) that lie in parallel with skeletal (extrafusal) muscle fibers. While extrafusal muscle fibers generate force via muscle contraction to initiate skeletal movement, muscle spindles are proprioceptive receptors in skeletal muscle that respond to the length of the muscle, thus playing an important role in motor control. Muscle spindle sensory output is sent to the CNS, where it is utilized to regulate motor activity through α-motor neurons signals to extrafusal skeletal muscle fibers, and to regulate proprioreceptor sensitivity through γ-motor neurons signals to intrafusal fibers [160]. Following peripheral nerve injury, structural changes occur in muscle spindles, which undergo atrophy [161]; given that a successful motor function recovery relies also on effective regeneration of this system, muscle spindles could be a promising experimental model to compare the regenerative capabilities of sensory and motor terminals, thus helping to fill the gap in knowledge on regeneration at sensory endings.

  1. Nascimento, A. I.; Mar, F. M.; Sousa, M. M., The intriguing nature of dorsal root ganglion neurons: Linking structure with polarity and function. Prog Neurobiol 2018, 168, 86-103.
  2. Zochodne, D. W., The challenges and beauty of peripheral nerve regrowth. J Periph Nerv Sys 2012, 17, 1–18.
  3. Toth, C. C.; Willis, D.; Twiss, J. L.; Walsh, S.; Martinez, J. A.; Liu, W. Q.; Midha, R.; Zochodne, D. W., Locally synthesized calcitonin gene-related Peptide has a critical role in peripheral nerve regeneration. J Neuropathol Exp Neurol 2009 68, 326–37.
  4. Höke, A.; Redett, R.; Hameed, H.; et al., Schwann cells express motor and sensory phenotypes that regulate axon regeneration. J Neurosci 2006, 26(38), 9646-55.
  5. Avraham, O; Deng, P. Y.; Jones, S; Kuruvilla, R.; Semenkovich, C. F.; Klyachko, V.A.; Cavalli, V., Satellite glial cells promote regenerative growth in sensory neurons. bioRxiv 2019, 12.13.874669
  6. Barton, M. J.; John, J. S.; Clarke, M.; Wright, A.; Ekberg, J., The Glia Response after Peripheral Nerve Injury: A Comparison between Schwann Cells and Olfactory Ensheathing Cells and Their Uses for Neural Regenerative Therapies. Int J Mol Sci 2017, 18(2), 287.
  7. Guo, X.; Colon, A.; Akanda, N.; et al., Tissue engineering the mechanosensory circuit of the stretch reflex arc with human stem cells: Sensory neuron innervation of intrafusal muscle fibers. Biomaterials 2017, 122, 179-187.
  8. Copray, J. C.; Brouwer, N., Neurotrophin‐3 mRNA expression in rat intrafusal muscle fibres after denervation and reinnervation. Neurosci Lett 1997, 236, 41‐ 4.
  • Reducing the amount of acronyms, which are numerous in several sections of the review. Amended in the revised manuscript.